# Deep Subspace Clustering Networks

**Pan Ji**[*]
University of Adelaide

**Tong Zhang**[*]
Australian National University

**Hongdong Li**
Australian National University

**Mathieu Salzmann**
EPFL - CVLab

**Ian Reid**
University of Adelaide

## Abstract

We present a novel deep neural network architecture for unsupervised subspace clustering. This architecture is built upon deep auto-encoders, which non-linearly map the input data into a latent space. Our key idea is to introduce a novel *self-expressive* layer between the encoder and the decoder to mimic the "self-expressiveness" property that has proven effective in traditional subspace clustering. Being differentiable, our new self-expressive layer provides a simple but effective way to learn pairwise affinities between all data points through a standard back-propagation procedure. Being nonlinear, our neural-network based method is able to cluster data points having complex (often nonlinear) structures. We further propose pre-training and fine-tuning strategies that let us effectively learn the parameters of our subspace clustering networks. Our experiments show that our method significantly outperforms the state-of-the-art unsupervised subspace clustering techniques.

## 1   Introduction

In this paper, we tackle the problem of subspace clustering [42] – a sub-field of unsupervised learning – which aims to cluster data points drawn from a union of low-dimensional subspaces in an unsupervised manner. Subspace clustering has become an important problem as it has found various applications in computer vision, e.g., image segmentation [50, 27], motion segmentation [17, 9], and image clustering [14, 10]. For example, under Lambertian reflectance, the face images of one subject obtained with a fixed pose and varying lighting conditions lie in a low-dimensional subspace of dimension close to nine [2]. Therefore, one can employ subspace clustering to group images of multiple subjects according to their respective subjects.

Most recent works on subspace clustering [49, 6, 10, 23, 46, 26, 16, 52] focus on clustering linear subspaces. However, in practice, the data do not necessarily conform to linear subspace models. For instance, in the example of face image clustering, reflectance is typically non-Lambertian and the pose of the subject often varies. Under these conditions, the face images of one subject rather lie in a non-linear subspace (or sub-manifold). A few works [5, 34, 35, 51, 47] have proposed to exploit the kernel trick [40] to address the case of non-linear subspaces. However, the selection of different kernel types is largely empirical, and there is no clear reason to believe that the implicit feature space corresponding to a predefined kernel is truly well-suited to subspace clustering.

In this paper, by contrast, we introduce a novel deep neural network architecture to learn (in an unsupervised manner) an explicit non-linear mapping of the data that is well-adapted to subspace clustering. To this end, we build our deep subspace clustering networks (*DSC-Nets*) upon deep auto-encoders, which non-linearly map the data points to a latent space through a series of encoder

---

[*]Authors contributed equally to this work

layers. Our key contribution then consists of introducing a novel *self-expressive* layer – a fully connected layer without bias and non-linear activations – at the junction between the encoder and the decoder. This layer encodes the "self-expressiveness" property [38, 9] of data drawn from a union of subspaces, that is, the fact that each data sample can be represented as a linear combination of other samples in the same subspace. To the best of our knowledge, our approach constitutes the first attempt to directly learn the affinities (through combination coefficients) between all data points within one neural network. Furthermore, we propose effective pre-training and fine-tuning strategies to learn the parameters of our DSC-Nets in an unsupervised manner and with a limited amount of data.

We extensively evaluate our method on face clustering, using the Extended Yale B [21] and ORL [39] datasets, and on general object clustering, using COIL20 [31] and COIL100 [30]. Our experiments show that our DSC-Nets significantly outperform the state-of-the-art subspace clustering methods.

## 2    Related Work

**Subspace Clustering.**    Over the years, many methods have been developed for linear subspace clustering. In general, these methods consist of two steps: the first and also most crucial one aims to estimate an affinity for every pair of data points to form an affinity matrix; the second step then applies normalized cuts [41] or spectral clustering [32] using this affinity matrix. The resulting methods can then be roughly divided into three categories [42]: factorization methods [7, 17, 44, 29, 16], higher-order model based methods [49, 6, 33, 37], and self-expressiveness based methods [9, 24, 26, 46, 15, 12, 22, 52]. In essence, factorization methods build the affinity matrix by factorizing the data matrix, and methods based on higher-order models estimate the affinities by exploiting the residuals of local subspace model fitting. Recently, self-expressiveness based methods, which seek to express the data points as a linear combination of other points in the same subspace, have become the most popular ones. These methods build the affinity matrix using the matrix of combination coefficients. Compared to factorization techniques, self-expressiveness based methods are often more robust to noise and outliers when relying on regularization terms to account for data corruptions. They also have the advantage over higher-order model based methods of considering connections between all data points rather than exploiting local models, which are often suboptimal. To handle situations where data points do not exactly reside in a union of linear subspaces, but rather in non-linear ones, a few works [34, 35, 51, 47] have proposed to replace the inner product of the data matrix with a pre-defined kernel matrix (e.g., polynomial kernel and Gaussian RBF kernel). There is, however, no clear reason why such kernels should correspond to feature spaces that are well-suited to subspace clustering. By contrast, here, we propose to explicitly learn one that is.

**Auto-Encoders.**    Auto-encoders (AEs) can non-linearly transform data into a latent space. When this latent space has lower dimension than the original one [13], this can be viewed as a form of non-linear PCA. An auto-encoder typically consists of an encoder and a decoder to define the data reconstruction cost. With the success of deep learning [20], deep (or stacked) AEs have become popular for unsupervised learning. For instance, deep AEs have proven useful for dimensionality reduction [13] and image denoising [45]. Recently, deep AEs have also been used to initialize deep embedding networks for unsupervised clustering [48]. A convolutional version of deep AEs was also applied to extract hierarchical features and to initialize convolutional neural networks (CNNs) [28].

There has been little work in the literature combining deep learning with subspace clustering. To the best of our knowledge, the only exception is [36], which first extracts SIFT [25] or HOG [8] features from the images and feeds them to a fully connected deep auto-encoder with a sparse subspace clustering (SSC) [10] prior. The final clustering is then obtained by applying k-means or SSC on the learned auto-encoder features. In essence, [36] can be thought of as a subspace clustering method based on k-means or SSC with deep auto-encoder features. Our method significantly differs from [36] in that our network is designed to directly learn the affinities, thanks to our new *self-expressive* layer.

## 3    Deep Subspace Clustering Networks (DSC-Nets)

Our deep subspace clustering networks leverage deep auto-encoders and the self-expressiveness property. Before introducing our networks, we first discuss this property in more detail.

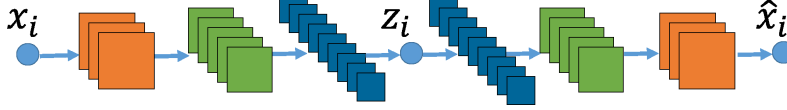

Figure 1: Deep Convolutional Auto-Encoder: The input $\mathbf{x}_i$ is mapped to $\mathbf{z}_i$ through an encoder, and then reconstructed as $\hat{\mathbf{x}}_i$ through a decoder. We use shaded circles to denote data vectors and shaded squares to denote the channels after convolution or deconvolution. We do not enforce the weights of the corresponding encoder and decoder layers to be coupled (or the same).

## 3.1 Self-Expressiveness

Given data points $\{\mathbf{x}_i\}_{i=1,\cdots,N}$ drawn from multiple linear subspaces $\{\mathcal{S}_i\}_{i=1,\cdots,K}$, one can express a point in a subspace as a linear combination of other points in the same subspace. In the literature [38, 9], this property is called self-expressiveness. If we stack all the points $\mathbf{x}_i$ into columns of a data matrix $\mathbf{X}$, the self-expressiveness property can be simply represented as one single equation, i.e., $\mathbf{X} = \mathbf{X}\mathbf{C}$, where $\mathbf{C}$ is the self-representation coefficient matrix. It has been shown in [15] that, under the assumption that the subspaces are independent, by minimizing certain norms of $\mathbf{C}$, $\mathbf{C}$ is guaranteed to have a block-diagonal structure (up to certain permutations), i.e., $c_{ij} \neq 0$ iff point $\mathbf{x}_i$ and point $\mathbf{x}_j$ lie in the same subspace. So we can leverage the matrix $\mathbf{C}$ to construct the affinity matrix for spectral clustering. Mathematically, this idea is formalized as the optimization problem

$$\min_{\mathbf{C}} \|\mathbf{C}\|_p \quad \text{s.t.} \quad \mathbf{X} = \mathbf{X}\mathbf{C}, \; (\text{diag}(\mathbf{C}) = \mathbf{0}) \;, \tag{1}$$

where $\| \cdot \|_p$ represents an arbitrary matrix norm, and the optional diagonal constraint on $\mathbf{C}$ prevents trivial solutions for sparsity inducing norms, such as the $\ell_1$ norm. Various norms for $\mathbf{C}$ have been proposed in the literature, e.g., the $\ell_1$ norm in Sparse Subspace Clustering (SSC) [9, 10], the nuclear norm in Low Rank Representation (LRR) [24, 23] and Low Rank Subspace Clustering (LRSC) [11, 43], and the Frobenius norm in Least-Squares Regression (LSR) [26] and Efficient Dense Subspace Clustering (EDSC) [15]. To account for data corruptions, the equality constraint in (1) is often relaxed as a regularization term, leading to

$$\min_{\mathbf{C}} \|\mathbf{C}\|_p + \frac{\lambda}{2}\|\mathbf{X} - \mathbf{X}\mathbf{C}\|_F^2 \quad \text{s.t.} \quad (\text{diag}(\mathbf{C}) = \mathbf{0}) \;. \tag{2}$$

Unfortunately, the self-expressiveness property only holds for linear subspaces. While kernel based methods [34, 35, 51, 47] aim to tackle the non-linear case, it is not clear that pre-defined kernels yield implicit feature spaces that are well-suited for subspace clustering. In this work, we aim to learn an explicit mapping that makes the subspaces more separable. To this end, and as discussed below, we propose to build our networks upon deep auto-encoders.

## 3.2 Self-Expressive Layer in Deep Auto-Encoders

Our goal is to train a deep auto-encoder, such as the one depicted by Figure 1, such that its latent representation is well-suited to subspace clustering. To this end, we introduce a new layer that encodes the notion of self-expressiveness.

Specifically, let $\Theta$ denote the auto-encoder parameters, which can be decomposed into encoder parameters $\Theta_e$ and decoder parameters $\Theta_d$. Furthermore, let $\mathbf{Z}_{\Theta_e}$ denote the output of the encoder, i.e., the latent representation of the data matrix $\mathbf{X}$. To encode self-expressiveness, we introduce a new loss function defined as

$$L(\Theta, \mathbf{C}) = \frac{1}{2}\|\mathbf{X} - \hat{\mathbf{X}}_\Theta\|_F^2 + \lambda_1\|\mathbf{C}\|_p + \frac{\lambda_2}{2}\|\mathbf{Z}_{\Theta_e} - \mathbf{Z}_{\Theta_e}\mathbf{C}\|_F^2 \quad \text{s.t.} \quad (\text{diag}(\mathbf{C}) = \mathbf{0}) \;, \tag{3}$$

where $\hat{\mathbf{X}}_\Theta$ represents the data reconstructed by the auto-encoder. To minimize (3), we propose to leverage the fact that, as discussed below, $\mathbf{C}$ can be thought of as the parameters of an additional network layer, which lets us solve for $\Theta$ and $\mathbf{C}$ jointly using backpropagation.[1]

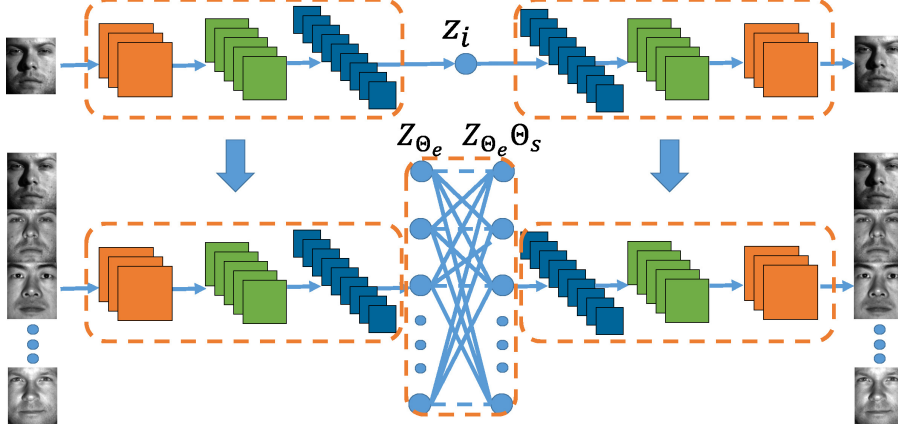

Figure 2: Deep Subspace Clustering Networks: As an example, we show a deep subspace clustering network with three convolutional encoder layers, one self-expressive layer, and three deconvolutional decoder layer. During training, we first pre-train the deep auto-encoder without the self-expressive layer; we then fine-tune our entire network using this pre-trained model for initialization.

Specifically, consider the self-expressiveness term in (3), $\|\mathbf{Z}_{\Theta_e} - \mathbf{Z}_{\Theta_e}\mathbf{C}\|_F^2$. Since each data point $\mathbf{z}_i$ (in the latent space) is approximated by a weighted linear combination of other points $\{\mathbf{z}_j\}_{j=1,\cdots,N}$ (optionally, $j \neq i$) with weights $c_{ij}$, this linear operation corresponds exactly to a set of linear neurons without non-linear activations. Therefore, if we take each $\mathbf{z}_i$ as a node in the network, we can then represent the self-expressiveness term with a fully-connected linear layer, which we call the *self-expressive layer*. The weights of the self-expressive layer correspond to the matrix $\mathbf{C}$ in (3), which are further used to construct affinities between all data points. Therefore, our self-expressive layer essentially lets us directly learn the affinity matrix via the network. Moreover, minimizing $\|\mathbf{C}\|_p$ simply translates to adding a regularizer to the weights of the self-expressive layer. In this work, we consider two kinds of regularizations on $\mathbf{C}$: (i) the $\ell_1$ norm, resulting in a network denoted by DSC-Net-L1; (ii) the $\ell_2$ norm, resulting in a network denoted by DSC-Net-L2.

For notational consistency, let us denote the parameters of the self-expressive layer (which are just the elements of $\mathbf{C}$) as $\Theta_s$. As can be seen from Figure 2, we then take the input to the decoder part of our network to be the transformed latent representation $\mathbf{Z}_{\Theta_e}\Theta_s$. This lets us re-write our loss function as

$$\tilde{L}(\tilde{\Theta}) = \frac{1}{2}\|\mathbf{X} - \hat{\mathbf{X}}_{\tilde{\Theta}}\|_F^2 + \lambda_1\|\Theta_s\|_p + \frac{\lambda_2}{2}\|\mathbf{Z}_{\Theta_e} - \mathbf{Z}_{\Theta_e}\Theta_s\|_F^2 \quad \text{s.t.} \quad (\text{diag}(\Theta_s) = \mathbf{0}), \quad (4)$$

where the network parameters $\tilde{\Theta}$ now consist of encoder parameters $\Theta_e$, self-expressive layer parameters $\Theta_s$, and decoder parameters $\Theta_d$, and where the reconstructed data $\hat{\mathbf{X}}$ is now a function of $\{\Theta_e, \Theta_s, \Theta_d\}$ rather than just $\{\Theta_e, \Theta_d\}$ in (3).

### 3.3 Network Architecture

Our network consists of three parts, i.e., stacked encoders, a self-expressive layer, and stacked decoders. The overall network architecture is shown in Figure 2. In this paper, since we focus on image clustering problems, we advocate the use of convolutional auto-encoders that have fewer parameters than the fully connected ones and are thus easier to train. Note, however, that fully-connected auto-encoders are also compatible with our self-expressive layer. In the convolutional layers, we use kernels with stride 2 in both horizontal and vertical directions, and rectified linear unit (ReLU) [19] for the non-linear activations. Given $N$ images to be clustered, we use all the images in a single batch. Each input image is mapped by the convolutional encoder layers to a latent vector (or node) $\mathbf{z}_i$, represented as a shaded circle in Figure 2. In the self-expressive layer, the nodes are fully connected using linear weights without bias and non-linear activations. The latent vectors are then mapped back to the original image space via the deconvolutional decoder layers.

For the $i^{\text{th}}$ encoder layer with $n_i$ channels of kernel size $k_i \times k_i$, the number of weight parameters is $k_i^2 n_{i-1} n_i$, with $n_0 = 1$. Since the encoders and decoders have symmetric structures, their total number of parameters is $\sum_i 2k_i^2 n_{i-1} n_i$ plus the number of bias parameters $\sum_i 2n_i - n_1 + 1$. For $N$ input images, the number of parameters for the self-expressive layer is $N^2$. For example, if we have

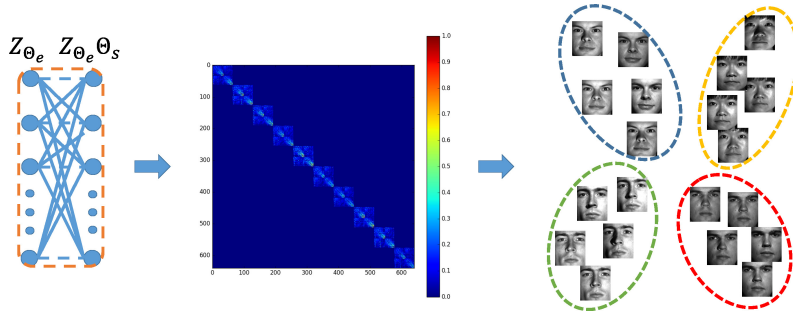

Figure 3: From the parameters of the self-expressive layer, we construct an affinity matrix, which we use to perform spectral clustering to get the final clusters. Best viewed in color.

three encoder layers with 10, 20, and 30 channels, respectively, and all convolutional kernels are of size $3 \times 3$, then the number of parameters for encoders and decoders is $\sum_{i=1}^{3} 2(k_i^2 n_{i-1} + 1)n_i - n_1 + 1 = 14671$. If we have 1000 input images, then the number of parameters in the self-expressive layer is $10^6$. Therefore, the network parameters are typically dominated by those of the self-expressive layer.

### 3.4 Training Strategy

Since the size of datasets for unsupervised subspace clustering is usually limited (e.g., in the order of thousands of images), our networks remain of a tractable size. However, for the same reason, it also remains difficult to directly train a network with millions of parameters from scratch. To address this, we design the pre-training and fine-tuning strategies described below. Note that this also allows us to avoid the trivial all-zero solution while minimizing the loss (4).

As illustrated in Figure 2, we first pre-train the deep auto-encoder without the self-expressive layer on all the data we have. We then use the trained parameters to initialize the encoder and decoder layers of our network. After this, in the fine-tuning stage, we build a big batch using all the data to minimize the loss $\tilde{L}(\Theta)$ defined in (4) with a gradient descent method. Specifically, we use Adam [18], an adaptive momentum based gradient descent method, to minimize the loss, where we set the learning rate to $1.0 \times 10^{-3}$ in all our experiments. Since we always use the same batch in each training epoch, our optimization strategy is rather a deterministic momentum based gradient method than a stochastic gradient method. Note also that, since we only have access to images for training and not to cluster labels, our training strategy is unsupervised (or self-supervised).

Once the network is trained, we can use the parameters of the self-expressive layer to construct an affinity matrix for spectral clustering [32], as illustrated in Figure 3. Although such an affinity matrix could in principle be computed as $|\mathbf{C}| + |\mathbf{C}^T|$, over the years, researchers in the field have developed many heuristics to improve the resulting matrix. Since there is no globally-accepted solution for this step in the literature, we make use of the heuristics employed by SSC [10] and EDSC [15]. Due to the lack of space, we refer the reader to the publicly available implementation of SSC and Section 5 of [15], as well as to the TensorFlow implementation of our algorithm [2] for more detail.

## 4 Experiments

We implemented our method in Python with Tensorflow-1.0 [1], and evaluated it on four standard datasets, i.e., the Extended Yale B and ORL face image datasets, and the COIL20/100 object image datasets. We compare our methods against the following baselines: Low Rank Representation (LRR) [23], Low Rank Subspace Clustering (LRSC) [43], Sparse Subspace Clustering (SSC) [10], Kernel Sparse Subspace Clustering (KSSC) [35], SSC by Orthogonal Matching Pursuit (SSC-OMP) [53], Efficient Dense Subspace Clustering (EDSC) [15], SSC with the pre-trained convolutional auto-encoder features (AE+SSC), and EDSC with the pre-trained convolutional auto-encoder features (AE+EDSC). For all the baselines, we used the source codes released by the authors and tuned the parameters by grid search to the achieve best results on each dataset. Since the code for the deep subspace clustering method of [36] is not publicly available, we are only able to provide a comparison

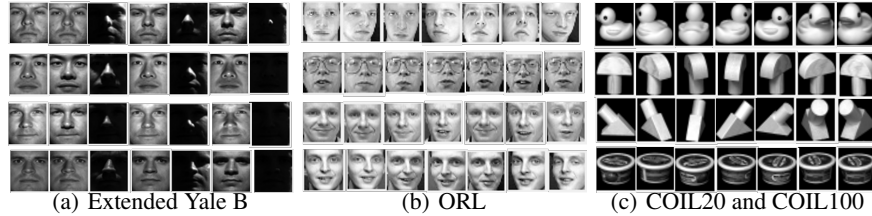

|                | (a) Extended Yale B | (b) ORL | (c) COIL20 and COIL100 |

Figure 4: Sample images from Extended Yale B, ORL , COIL20 and COIL100.

| layers | encoder-1 | encoder-2 | encoder-3 | self-expressive | decoder-1 | decoder-2 | decoder-3 |
|---|---|---|---|---|---|---|---|
| kernel size | $5 \times 5$ | $3 \times 3$ | $3 \times 3$ | – | $3 \times 3$ | $3 \times 3$ | $5 \times 5$ |
| channels | 10 | 20 | 30 | – | 30 | 20 | 10 |
| parameters | 260 | 1820 | 5430 | 5914624 | 5420 | 1810 | 251 |

Table 1: Network settings for Extended Yale B.

against this approach on Extended Yale B and COIL20, for which the results are provided in [36]. Note that this comparison already clearly shows the benefits of our approach.

For all quantitative evaluations, we make use of the clustering error rate, defined as

$$\text{err } \% = \frac{\text{\# of wrongly clustered points}}{\text{total \# of points}} \times 100\% \ . \tag{5}$$

## 4.1 Extended Yale B Dataset

The Extended Yale B dataset [21] is a popular benchmark for subspace clustering. It consists of 38 subjects, each of which is represented with 64 face images acquired under different illumination conditions (see Figure 4(a) for sample images from this dataset). Following the experimental setup of [10], we down-sampled the original face images from $192 \times 168$ to $42 \times 42$ pixels, which makes it computationally feasible for the baselines [10, 23]. In each experiment, we pick $K \in \{10, 15, 20, 25, 30, 35, 38\}$ subjects (each subject with 64 face images) to test the robustness w.r.t. an increasing number of clusters. Taking all possible combinations of $K$ subjects out of 38 would result in too many experimental trials. To get a manageable size of experiments, we first number the subjects from 1 to 38 and then take all possible $K$ consecutive subjects. For example, in the case of 10 subjects, we take all the images from subject $1 - 10, 2 - 11, \cdots, 29 - 38$, giving rise to 29 experimental trials.

We experimented with different architectures for the convolutional layers of our network, e.g., different network depths and number of channels. While increasing these values increases the representation power of the network, it also increases the number of network parameters, thus requiring larger training datasets. Since the size of Extended Yale B is quite limited, with only 2432 images, we found having three-layer encoders and decoders with $[10, 20, 30]$ channels to be a good trade-off for this dataset. The detailed network settings are described in Table 1. In the fine-tuning phase, since the number of epochs required for gradient descent increases as the number of subjects $K$ increases, we defined the number of epochs for DSC-Net-L1 as $160 + 20K$ and for DSC-Net-L2 as $50 + 25K$. We set the regularization parameters to $\lambda_1 = 1.0, \lambda_2 = 1.0 \times 10^{\frac{K}{10} - 3}$.

The clustering performance of different methods for different numbers of subjects is provided in Table 2. For the experiments with $K$ subjects, we report the mean and median errors of $39 - K$ experimental trials. From these results, we can see that the performance of most of the baselines decreases dramatically as the number of subjects $K$ increases. By contrast, the performance of our deep subspace clustering methods, DSC-Net-L1 and DSC-Net-L2, remains relatively stable w.r.t. the number of clusters. Specifically, our DSC-Net-L2 achieves $2.67\%$ error rate for 38 subjects, which is only around $1/5$ of the best performing baseline EDSC. We also observe that using the pre-trained auto-encoder features does not necessarily improve the performance of SSC and EDSC, which confirms the benefits of our joint optimization of all parameters in one network. The results of [36] on this dataset for 38 subjects was reported to be $92.08 \pm 2.42\%$ in terms of clustering accuracy, or equivalently $7.92 \pm 2.42\%$ in terms of clustering error, which is worse than both our methods – DSC-Net-L1 and DSC-Net-L2. We further notice that DSC-Net-L1 performs slightly worse than DSC-Net-L2 in the current experimental settings. We conjecture that this is due to the difficulty in optimization introduced by the $\ell_1$ norm as it is non-differentiable at zero.

| Method | LRR | LRSC | SSC | AE+SSC | KSSC | SSC-OMP | EDSC | AE+EDSC | DSC-Net-L1 | DSC-Net-L2 |
|---|---|---|---|---|---|---|---|---|---|---|
| **10 subjects** | | | | | | | | | | |
| Mean | 22.22 | 30.95 | 10.22 | 17.06 | 14.49 | 12.08 | 5.64 | 5.46 | 2.23 | **1.59** |
| Median | 23.49 | 29.38 | 11.09 | 17.75 | 15.78 | 8.28 | 5.47 | 6.09 | 2.03 | **1.25** |
| **15 subjects** | | | | | | | | | | |
| Mean | 23.22 | 31.47 | 13.13 | 18.65 | 16.22 | 14.05 | 7.63 | 6.70 | 2.17 | **1.69** |
| Median | 23.49 | 31.64 | 13.40 | 17.76 | 17.34 | 14.69 | 6.41 | 5.52 | 2.03 | **1.72** |
| **20 subjects** | | | | | | | | | | |
| Mean | 30.23 | 28.76 | 19.75 | 18.23 | 16.55 | 15.16 | 9.30 | 7.67 | 2.17 | **1.73** |
| Median | 29.30 | 28.91 | 21.17 | 16.80 | 17.34 | 15.23 | 10.31 | 6.56 | 2.11 | **1.80** |
| **25 subjects** | | | | | | | | | | |
| Mean | 27.92 | 27.81 | 26.22 | 18.72 | 18.56 | 18.89 | 10.67 | 10.27 | 2.53 | **1.75** |
| Median | 28.13 | 26.81 | 26.66 | 17.88 | 18.03 | 18.53 | 10.84 | 10.22 | 2.19 | **1.81** |
| **30 subjects** | | | | | | | | | | |
| Mean | 37.98 | 30.64 | 28.76 | 19.99 | 20.49 | 20.75 | 11.24 | 11.56 | 2.63 | **2.07** |
| Median | 36.82 | 30.31 | 28.59 | 20.00 | 20.94 | 20.52 | 11.09 | 10.36 | 2.81 | **2.19** |
| **35 subjects** | | | | | | | | | | |
| Mean | 41.85 | 31.35 | 28.55 | 22.13 | 26.07 | 20.29 | 13.10 | 13.28 | 3.09 | **2.65** |
| Median | 41.81 | 31.74 | 29.04 | 21.74 | 25.92 | 20.18 | 13.10 | 13.21 | 3.10 | **2.64** |
| **38 subjects** | | | | | | | | | | |
| Mean | 34.87 | 29.89 | 27.51 | 25.33 | 27.75 | 24.71 | 11.64 | 12.66 | 3.33 | **2.67** |
| Median | 34.87 | 29.89 | 27.51 | 25.33 | 27.75 | 24.71 | 11.64 | 12.66 | 3.33 | **2.67** |

Table 2: Clustering error (in %) on Extended Yale B. The lower the better.

| layers | encoder-1 | encoder-2 | encoder-3 | self-expressive | decoder-1 | decoder-2 | decoder-3 |
|---|---|---|---|---|---|---|---|
| kernel size | $5 \times 5$ | $3 \times 3$ | $3 \times 3$ | – | $3 \times 3$ | $3 \times 3$ | $5 \times 5$ |
| channels | 5 | 3 | 3 | – | 3 | 3 | 5 |
| parameters | 130 | 138 | 84 | 160000 | 84 | 140 | 126 |

Table 3: Network settings for ORL.

## 4.2 ORL Dataset

The ORL dataset [39] is composed of 400 human face images, with 40 subjects each having 10 samples. Following [4], we down-sampled the original face images from $112 \times 92$ to $32 \times 32$. For each subject, the images were taken under varying lighting conditions with different facial expressions (open / closed eyes, smiling / not smiling) and facial details (glasses / no glasses)(see Figure 4(b) for sample images). Compared to Extended Yale B, this dataset is more challenging for subspace clustering because (i) the face subspaces have more non-linearity due to varying facial expressions and details; (ii) the dataset size is much smaller (400 vs. 2432). To design a trainable deep auto-encoder on 400 images, we reduced the number of network parameters by decreasing the number of channels in each encoder and decoder layer. The resulting network is specified in Table 3.

Since we already verified the robustness of our method to the number of clusters in the previous experiment, here, we only provide results for clustering all 40 subjects. In this setting, we set $\lambda_1 = 1$ and $\lambda_2 = 0.2$ and run 700 epochs for DSC-Net-L2 and 1500 epochs for DSC-Net-L1 during fine-tuning. Note that, since the size of this dataset is small, we can even use the whole data as a single batch in pre-training. We found this to be numerically more stable and converge faster than stochastic gradient descent using randomly sampled mini-batches.

Figure 5(a) shows the error rates of the different methods, where different colors denote different subspace clustering algorithms and the length of the bars reflects the error rate. Since there are much fewer samples per subject, all competing methods perform worse than on Extended Yale B. Note that both EDSC and SSC achieve moderate clustering improvement by using the features of pre-trained convolutional auto-encoders, but their error rates are still around twice as high as those of our methods.

## 4.3 COIL20 and COIL100 Datasets

The previous experiments both target face clustering. To show the generality of our algorithm, we also evaluate it on the COIL object image datasets – COIL20 [31] and COIL100 [30]. COIL20 consists of 1440 gray-scale image samples, distributed over 20 objects such as duck and car model (see sample images in Figure 4(c)). Similarly, COIL100 consists of 7200 images distributed over 100 objects. Each object was placed on a turntable against a black background, and 72 images were taken at pose intervals of 5 degrees. Following [3], we down-sampled the images to $32 \times 32$. In contrast with the previous human face datasets, in which faces are well aligned and have similar structures, the object images from COIL20 and COIL100 are more diverse, and even samples from

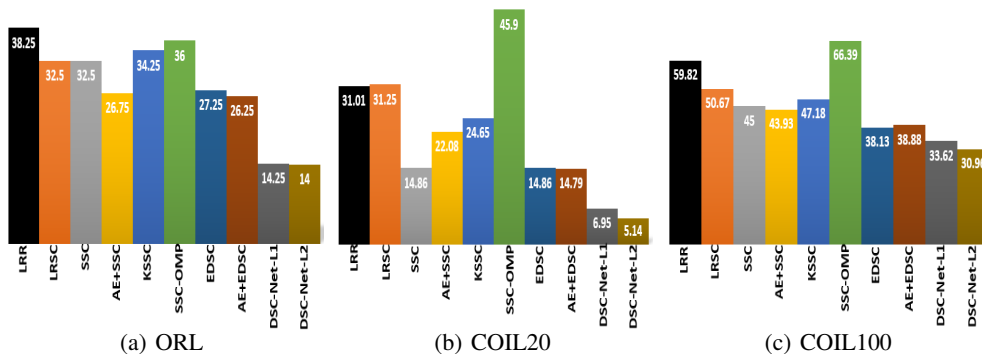

|  | ORL | COIL20 | COIL100 |
|---|---|---|---|

(a) ORL          (b) COIL20          (c) COIL100

Figure 5: Subspace clustering error (in %) on the ORL, COIL20 and COIL100 datasets. Different colors indicate different methods. The height of the bars encodes the error, so the lower the better.

| | COIL20 | | | COIL100 | | |
|---|---|---|---|---|---|---|
| layers | encoder-1 | self-expressive | decoder-1 | encoder-1 | self-expressive | decoder-1 |
| kernel size | $3 \times 3$ | – | $3 \times 3$ | $5 \times 5$ | – | $5 \times 5$ |
| channels | 15 | – | 15 | 50 | – | 50 |
| parameters | 150 | 2073600 | 136 | 1300 | 51840000 | 1251 |

Table 4: Network settings for COIL20 and COIL100.

the same object differ from each other due to the change of viewing angle. This makes these datasets challenging for subspace clustering techniques. For these datasets, we used shallower networks with one encoder layer, one self-expressive layer, and one decoder layer. For COIL20, we set the number of channels to 15 and the kernel size to $3 \times 3$. For COIL100, we increased the number of channels to 50 and the kernel size to $5 \times 5$. The settings for both networks are provided in Table 4. Note that with these network architectures, the dimension of the latent space representation $z_i$ increases by a factor of 15/4 for COIL20 (as the spatial resolution of each channel shrinks to 1/4 of the input image after convolutions with stride 2, and we have 15 channels) and 50/4 for COIL100. Thus our networks perform dimensionality lifting rather than dimensionality reduction. This, in some sense, is similar to the idea of Hilbert space mapping in kernel methods [40], but with the difference that, in our case, the mapping is explicit, via the neural network. In our experiments, we found that these shallow, dimension-lifting networks performed better than deep, bottle-neck ones on these datasets. While it is also possible to design deep, dimension-lifting networks, the number of channels has to increase by a factor of 4 after each layer to compensate for the resolution loss. For example, if we want the latent space dimension to increase by a factor of 15/4, we need $15 \cdot 4$ channels in the second layer for a 2-layer encoder, $15 \cdot 4^2$ channels in the third layer for a 3-layer encoder, and so forth. In the presence of limited data, this increasing number of parameters makes training less reliable. In our fine-tuning stage, we ran 30 epochs (COIL20) / 100 epochs (COIL100) for DSC-Net-L1 and 30 epochs (COIL20) / 120 epochs (COIL100) for DSC-Net-L2, and set the regularization parameters to $\lambda_1 = 1, \lambda_2 = 150/30$ (COIL20/COIL100).

Figure 5(b) and (c) depict the error rates of the different methods on clustering 20 classes for COIL20 and 100 classes for COIL100, respectively. Note that, in both cases, our DSC-Net-L2 achieves the lowest error rate. In particular, for COIL20, we obtain an error of 5.14%, which is roughly 1/3 of the error rate of the best-performing baseline AE+EDSC. The results of [36] on COIL20 were reported to be $14.24 \pm 4.70\%$ in terms of clustering error, which is also much higher than ours.

## 5   Conclusion

We have introduced a deep auto-encoder framework for subspace clustering by developing a novel self-expressive layer to harness the "self-expressiveness" property of a union of subspaces. Our deep subspace clustering network allows us to directly learn the affinities between all data points with a single neural network. Furthermore, we have proposed pre-training and fine-tuning strategies to train our network, demonstrating the ability to handle challenging scenarios with small-size datasets, such as the ORL dataset. Our experiments have demonstrated that our deep subspace clustering methods provide significant improvement over the state-of-the-art subspace clustering solutions in terms of clustering accuracy on several standard datasets.

**Acknowledgements**

This research was supported by the Australian Research Council (ARC) through the Centre of Excellence in Robotic Vision, CE140100016, and through Laureate Fellowship FL130100102 to IDR. TZ was supported by the ARC's Discovery Projects funding scheme (project DP150104645).

## Footnotes

[1]Note that one could also alternate minimization between $\Theta$ and $\mathbf{C}$. However, since the loss is non-convex, this would not provide better convergence guarantees and would require investigating the influence of the number of steps in the optimization w.r.t. $\Theta$ on the clustering results.

[2]https://github.com/panji1990/Deep-subspace-clustering-networks

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
