[Reviews · NeurIPS 2017]

Reviewer 1



The paper propose to enforce self-expressiveness within the latent space of an auto-encoder, so as to make the latent representation better suited to spectral clustering. The underlying intuition, and the experimental validation sound convincing (comparison with key relevant alternative methods is provided). Paper is also clear and well writen. I do encourage its acceptance. Some aspects however deserve a deeper discussion (even if they point out relative weaknesses of the approach). These include the practical use of the algorithm: how does the algorithm deal with large datasets (the number of parameters in the model increases with the number N of samples)? how can you assign a new sample to one of the cluster (without running the algorithm on the whole dataset)? what is the sensitivity of the approach to \lambda_2? what does motivate the definition of lambda_2 on page 6 ?

Reviewer 2



The paper addresses the problem of subspace clustering, i.e., separating a collection of data points lying in a union of subspaces according to underlying subspaces, using deep neural networks. To do so, the paper builds on the sparse subspace clustering idea: among all possible representations of a data point as a combination of other points in the dataset, the representation that uses the minimum number of points, corresponds to points from the same subspace. In other words, SSC uses the idea that for a data matrix $X$, a sparse solution of $X = X C$ (subject to $diag(C) = 0$) represents each point as a combination of a few other points from the same subspace. The paper proposes a deep neural network to transform the data into a new representation $Z = f_W(X)$ for which one searches for a sparse representation of $Z = Z C$, with the hope to learn more effective representations of data for clustering. To achieve this, the paper uses an auto-encoder scheme, where the middle hidden layer outputs are used as $Z$. To enforce $Z - Z C = 0$, the paper repeats the middle layer, with the hope that these two consecutive layers obtain same activations. If so, the weights between the two layers would correspond to coefficient matrix $C$ which will be used then, similar to existing methods, to build an affinity matrix on data and to obtain clustering. The paper demonstrates experiments on Extended YaleB and ORL face datasets as well as COIL20/100. The reviewer believes that the paper is generally well-written and easy to read. The paper also takes a sufficiently interesting approach to connect subspace clustering with deep learning. On the other hand, there are several issues with the technical approach and the experiments that limit the novelty and correctness of the paper. 1) The reviewer believes that the idea of the paper is similar to the one in [36], which uses an auto-encoder scheme, taking the middle layer as $Z$. While [36] fixes the $C$ and solves for $Z$, the proposed method repeats the middle layer to also solve for $C$. Despite this similarity, these is only a sentence in the paper (line 82) referring to this work and the experiments do not compare against [36]. Thus there is a need to clearly discuss advantages and disadvantages with respect to [36] and compare against it in the experiments. 2) It is not clear to the reviewer how the paper enforces that the activations of the middle and the middle plus one layer must be the same. While, clearly one can initialize $C$ so that the consecutive activations will be the same, one the back propagation is done and the weights of the auto-encoder change, the activations of middle layer and the one after it will not be the same (unless the $\lambda_2$ is very large and the network achieves it global minima, which is generally not the case). As a result, in the next iteration, the network will minimize $Z - Y C$ for different $Y$ and $Z$, which is not desired. 3) One of the drawbacks of the proposed approach is the fact that the number of weights between the middle layers of the network is $N^2$, given $N$ data points. Thus, it seems that the method will not be scalable to very large datasets. 4) In the experiments (line 243), the paper discusses training using mini-batches. Notice that while using mini-batches makes sense in the context of recognition, where each sample can be applied separately to the network, using mini-batches does not make sense in the context of clustering discussed in the paper. Using a mini-batch of size $M < N$, how does the paper train a neural network between two layers, each with $N$ nodes? In other words, it is not clear how the mini-batch instances and in what order will be input the the network.

Reviewer 3



This paper proposes a new subspace clustering (SC) method based on neural networks. Specifically, the authors constructed the network by adding a self-expressive layer to the latent space of the traditional auto-encoder (AE) network, and used the coefficients of the self-expression to compute the affinity matrix for the final clustering. The idea of doing SC in the latent space of AE is reasonable since the features may be more powerful for clustering task than the original data. Besides, the designing of the self-expressive layer is able to further exploit the structure of the data in latent space. The experiments on three image datasets demonstrated the effectiveness of the proposed method. My concerns are as follows: First, as indicated by the authors, [36] also used AE type networks for SC, and thus it is better to discuss the relations and differences of network structures between the two in details. It would also be more convincing to directly compare the two methods in experiments, instead of saying the code "is publicly available" and only including partial results reported in [36]. Second, since the hyperparameters are generally very important for deep learning, it would be better to discuss how the hyperparameters, e.g., the number of layers, influence the final performance. Third, it would also be useful to include the computation time for readers who are willing to follow this work.